# Effects of Biodegradable Liquid Film on the Soil and Fruit Quality of *Vitis Franco-american* L. Hutai-8 Berries

Xinyao Duan [1], Yasai Yan [1], Xing Han [1], Ying Wang [1], Rihui Li [1], Feifei Gao [1], Liang Zhang [1], Ruteng Wei [1], Hua Li [1,2,3,4,*] and Hua Wang [1,2,3,4,*]

[1] College of Enology, Northwest A&F University, Yangling, Xianyang 712100, China; duanxinyao@nwafu.edu.cn (X.D.); yanyasai@nwafu.edu.cn (Y.Y.); hanxing@nwafu.edu.cn (X.H.); wangying2018@nwafu.edu.cn (Y.W.); 2021056184@nwafu.edu.cn (R.L.); gaofeifei@nwafu.edu.cn (F.G.); zhangliang20@nwafu.edu.cn (L.Z.); weiruteng2019@nwafu.edu.cn (R.W.)

[2] Engineering Research Center for Viti-Viniculture, National Forestry and Grassland Administration, Yangling, Xianyang 712100, China

[3] Shaanxi Engineering Research Center for Viti-Viniculture, Yangling, Xianyang 712100, China

[4] China Wine Industry Technology Institute, Yinchuan 750000, China

* Correspondence: lihuawine@nwafu.edu.cn (H.L.); wanghua@nwafu.edu.cn (H.W.); Tel.: +86-29-87082805 (H.L.)

**Abstract:** Biodegradable liquid mulch film (LF), which can be degraded naturally without harming the environment, is a new type of covering material that provides an environmentally friendly alternative to plastic mulch film (PF). In this study, the effects of LF and PF (ploughing (CK) used as a control) on the soil and fruit quality of Hutai-8 were evaluated through an experiment, and several soil physicochemical properties and indicators of fruit quality were measured. In-row mulching significantly increased the content of total potassium, available phosphorus, and available potassium in the topsoil (0–20 cm), the ripeness of the grape berries, and the content of phenolics in the skin. The effects were consistent between the two years. The effect of LF was more pronounced in the same year, indicating that LF is an effective alternative to PF. Therefore, LF can be used as an environmentally friendly substitute for PF to improve soil and fruit quality and incorporated into cultivation management plans. Correlation analysis revealed that the content of reducing sugars, flavonoids, total phenols, flavan-3-ols, and anthocyanins, as well as fruit ripeness, increased as the content of total potassium, available phosphorus, and available potassium in the soil increased.

**Keywords:** biodegradable liquid film; sustainable; soil nutrients; berry quality

## 1. Introduction

Grapes are an important crop in China with high nutritional benefits [1]. Soil quality is closely related to grape yield and quality [2]. The roots of grapevines are generally distributed in the soil at a depth of 15–80 cm, mostly concentrated at 20–40 cm, and the deepest can reach more than 1 m [3–6]. The distribution of roots in the soil is greatly affected by the soil type, texture, water, nutrients, and the growth and development of the shoots. The growth of the roots is water-oriented, fertilizer-oriented, and geotropic. The deeper the fertilization, the deeper the root system is, and vice versa [7–9]. In orchards, higher soil temperatures and soil water content in early spring allow grape rhizomes to develop earlier and enhance their activity [10]. This promotes the early growth of grape branches and leaves, as well as photosynthesis [10]. Grape quality and soil quality can be improved by altering cultivation practices.

Orchard mulching has been shown to be a green and sustainable orchard management model [11], as it can regulate soil temperature and humidity, protect the soil from erosion [12–14], reduce surface runoff, prevent nutrient loss [15], improve soil fertility [16,17], alter soil structure [18], reduce water evaporation, maintain soil moisture, control the spread

of weeds in gardens, improve soil microbial structure and functional diversity [19], and enhance the reproduction of microbes [20–22]. Mulching cultivation technology involves using different materials to cover the surface of agricultural fields to improve the temperature and moisture level of soil and control the spread of weeds [23–26]. Mulching treatment can improve the water retention capacity of soil, increase soil enzyme activity [27], improve soil structure, significantly reduce soil volume weight, increase soil porosity, improve soil permeability (which benefits root growth) [28], optimize the development of shallow and lateral roots, and increase root density and growth [29]. It can also increase the content of soil nutrients in the short term [30–33] and increase the soil carbon–nitrogen ratio [26]. Mulching treatment results in the advancement of grape phenology, increases the fruit setting rate, and makes fruit mature earlier [10]. It can also increase the chlorophyll content of grape leaves [34,35], enhance the rate of photosynthesis and rate of color change of grapes, and promote fruit maturity [31], all of which increase fruit yield and quality. Mulching, of course, can also bring negative effects. Previous studies have shown that short-term surface mulching is beneficial to improve soil microbial activity, but long-term plastic film mulching deteriorates the soil's physical and chemical properties, which is not conducive to soil microbial activity [36,37]. In addition, the soil organic carbon content of long-term plastic film mulching is continuously reduced [38].

Some industrialized countries have begun to use plastic film (PF) as a ground cover for the cultivation of crops, given the wide availability of plastics. PF can promote water conservation [39], heat preservation [40], and improve crop yields [41–43]. Polyethylene mulches have been widely used in agriculture for over half a century [44,45]. Most PFs are made of low-density polyethylene that requires at least several hundred years to completely degrade in soil [46,47]. The long-term continuous use of PF results in the deposition of residual PF, and thus irreversible pollution of the soil [45,48,49]. In addition, residual PFs disintegrate into microplastics smaller than 5 nm in diameter [50], which reduces the health of the environment [51]. Plastic residues can have a significant negative impact on the soil environment and agricultural production when they are excessive; for example, they can result in the destruction of soil structure [52], reductions in soil moisture infiltration [53], delayed soil water and nutrient movement [53,54], root growth inhibition [55,56], and reductions in crop yields [52,54,57]. Alternative materials are needed to replace traditional PFs and resolve the problems associated with residual plastic pollution [58].

Biodegradable liquid mulch film (LF) is considered an excellent substitute for PF, and it has been used in several countries, such as Norway [59], Japan [60], China [61,62], and Spain [63], to prevent pollution from plastic residues. There is growing interest in the development of LF for mulching crops to minimize the environmental impacts of polyethylene films [64]. Numerous studies have shown that soil temperatures and the soil water content under LF mulching are similar to those under PF in the early growth stage, slightly decreased during the middle stage of crop growth due to the partial degradation of the LF, and significantly decreased during the late stage of crop growth due to considerable degradation of the LF [61,62,64–66]. The average soil temperature under biodegradable mulch is 2–3 °C lower at a 10 cm soil depth compared with that under PF in the late stage of crop growth. The water storage capacity is also reduced when the degradation rate of LF is rapid [67]. However, differences in soil microbial biomass and enzyme activities between LF and PF were minor, especially differences in soil moisture and the nitrogen content [68].

LF can be used as a binder for stabilizing soil aggregate structure, which can connect soil particles to form agglomerates, reduce surface damage and soil wind erosion, improve soil structure, regulate the physical and chemical properties of the soil, promote crop growth and development, promote the growth and reproduction of microorganisms, promote the transformation and accumulation of soil organic matter, and improve soil fertility [69,70]. The LF used in this experiment was developed as an environmentally friendly soil structure conditioner that is highly adhesive and can quickly form a multi-molecular network film after it is sprayed on the soil surface, which closes the pores on the soil surface and minimizes the evaporation of soil moisture without affecting the infiltration of precipitation.

Covering LF is a new moisture-preserving cultivation technology that can be naturally degraded under the action of sunlight and soil microorganisms and does not damage the ecological environment [71–73].

In this study, the effects of LF on soil properties and the fruit quality of Hutai No. 8 were investigated by measuring various soil physicochemical properties and fruit quality indicators for two consecutive years. The efficacy of LF for improving soil and berry quality was examined relative to CK and PF mulching.

## 2. Materials and Methods

### 2.1. Grapevine Field Conditions

The experiment was carried out in a flat orchard in Shengtang Winery in the middle of the Guanzhong Plain, Yangling, Shaanxi, China (34°27′ N; 108°8′ E) in 2019 and 2020. The site has a typical semi-humid and semi-arid climate of the warm temperate zone of East Asia; the altitude is 514 m, the mean annual temperature is 15.4 °C, the mean annual rainfall is 660 mm (data from a meteorological weather station in Jinghe, China), the frost-free period is 211 d, the annual average sunshine duration is 2163.8 h, and the annual total solar radiation is 480.79 kJ/cm$^2$ (data from China Statistical Yearbook). Values of monthly mean temperature and monthly precipitation (from January to December in 2019 and 2020) are shown in Figure 1. Soils in the orchard were relatively uniform and mainly loam [74].

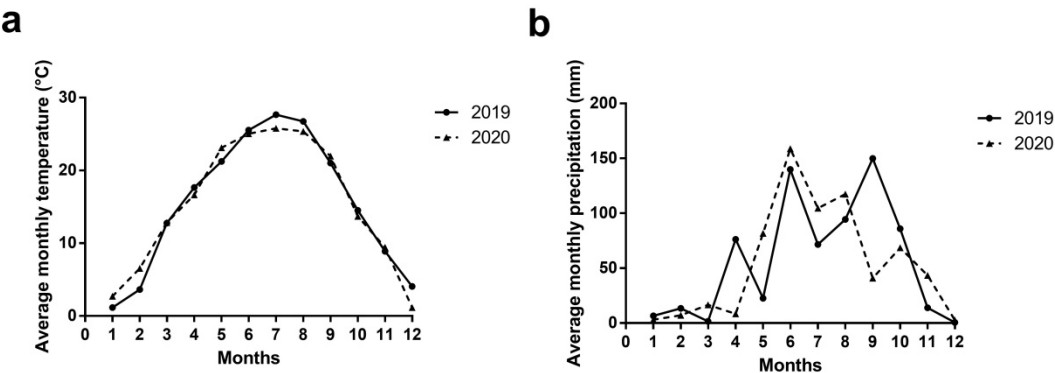

**Figure 1.** Average monthly temperature (**a**) and precipitation (**b**) obtained during 2019 and 2020. Notes: Data from China Statistical Yearbook.

Hutai-8 (*Vitis Franco-american* L.) plants in the orchard were planted in 2008, with a row spacing of 1.0 m × 2.5 m and a row length of 90 m. Plants were arranged in a single hedge frame system, and single stems and double arms with long and short branches were mixed and trimmed. Hutai-8 is a European and American hybrid that was bred by the Xi'an Grape Research Institute through the 'Olympia' bud mutation. The variety is resistant to disease and drought. The shoots are green and shiny. The leaves are nearly round. Flowers are bisexual. The ear is 30 cm long, 18 cm wide, conical with a secondary ear. The ear and the grain are tight. The top of the fruit is black-purple, the tail is purple-red, and the fruit powder is thick.

### 2.2. Treatment and Sampling

The field trials were carried out during the 2019 and 2020 growing season. The in-row mulching treatment was carried out before germination in March each year. The three treatments were as follows: (i) biodegradable liquid mulch film (LF); (ii) plastic film (PF); and (iii) control (CK). A single factor level block design with three replicates was used. The experimental plot was divided into three blocks with three replicates, two rows per treatment, and 90 vines per row. To avoid wind drift and edge effects, the treatments were arranged in nonadjacent rows. The figure of each block as replicate is shown in Figure 2. The experimental setup had three such blocks. In the CK plot, the soil was ploughed and re-ploughed until harvest. In the PF plot, soil was covered by a 1-m wide silver-black

two-color PF made by Hongdae Plastic Factory. In the LF plot, soil was covered by LF purchased from Shaanxi Kerui Company. LF was sprayed into rows using a knapsack sprayer and re-sprayed until harvest as needed. Soil sprayed with LF is black and becomes lighter as LF degrades.

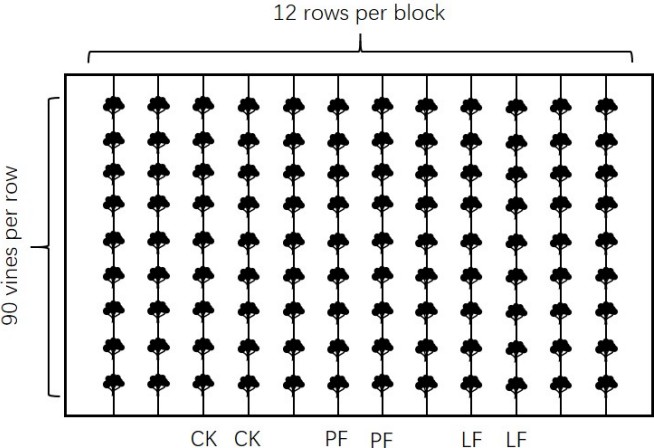

**Figure 2.** The experimental setup of each block as replicate. Notes: Two rows per treatment. A total of 90 vines per row. The treatments were arranged in nonadjacent rows. The experimental setup had three such blocks.

Soil samples were collected during the fruit harvesting period using the 5-point sampling method in the three treatments in 2019 and 2020. The 0–20 cm and 20–40 cm soil samples were collected with a 4 cm diameter soil drill. Five soil samples collected in the same treatment were mixed; after removing small rocks, animal and plant residues, and roots, they were sealed and brought to the laboratory at a low temperature for packaging. The soil was air-dried and placed through a 0.15-mm sieve for determination of the total nutrient, total carbon, and total organic matter content of the soil and through a 1-mm sieve for determination of the available nutrients and soil pH [75]. All measurements were taken three times, and all parameters were analyzed in triplicate.

Berry samples were collected during the fruit harvesting period, using the "Z"-shaped sampling method in the three treatments in 2019 and 2020. Fifty vines were selected for each treatment, one ear of grapes was randomly selected from each tree, and 25 ears were selected on the shaded side and sunny side. One grape was randomly selected from the top and bottom, left and right, and front and back of each ear of grapes; thus, a total of six grapes were collected from each ear of grapes, and a total of 300 grapes were collected for each treatment [1]. All measurements were taken three times, and all parameters were analyzed in triplicate.

*2.3. Soil Analyses*

2.3.1. Total Nutrient Content

Measurements of total nitrogen, total phosphorus, and total potassium were made following the methods of [76] and [77], using the $H_2SO_4$-$H_2O_2$ digestion method with modifications. Air-dried soil samples (0.3–0.5 g) were passed through a 0.15-mm steel sieve into a 50-mL digestion tube; 5 mL of concentrated $H_2SO_4$ was added, and the mixture was shaken well overnight. Next, the tube was digested at 150 °C for 0.5 h and 380 °C for 1 h; it was then cooled, and 2 mL of $H_2O_2$ was added for digestion at 260 °C for 10 min. The mixture was cooled, and this procedure was repeated four times. Thus, the total amount of $H_2O_2$ added was 8–10 mL. After cooling, the digestion liquid was transferred to a 100-mL volumetric flask, shaken well overnight, and stored in a refrigerator at 4 °C. Ten ml of the supernatant was placed into a 10-mL centrifuge tube to determine the total nitrogen and total potassium content. Two ml of the supernatant was placed into a 10-mL centrifuge

tube, and 1 mL of 4 mol/L NaOH and 7 mL of water were added to determine the total phosphorus (TP) content. The content of total nitrogen (TN) and TP was measured by an AA3 continuous flow analyzer, and the content of total potassium (TK) was measured by a flame photometer.

### 2.3.2. Available Nutrient Content

Measurements of available nutrients were based on the methods of [78] with some modifications. Air-dried soil samples (5.000 g) were passed through a 1-mm steel sieve into a 150-mL conical flask; 50 mL of 1 mol/L KCl was added for extraction in a shaker at 120 r/min for 1 h. The mixture was then filtered and stored in a refrigerator at 4 °C for subsequent determination of the available nitrogen (AN) content. To determine the available phosphorus (AP) content, air-dried soil samples (2.500 g) were passed through a 1-mm steel sieve into a 150-mL conical flask, and 1 g of phosphorus-free activated carbon and 50 mL of 0.5 mol/L NaHCO$_3$ were added for extraction in a shaking table at 120 r/min for 30 min. After letting the mixture stand for 30 min, the mixture was filtered, 1:1 neutralized with 0.5 mol/LHCl, and stored in a refrigerator at 4 °C for subsequent determination of the AP content. To determine the available potassium (AK) content, air-dried soil samples (5.000 g) were passed through a 1-mm steel sieve into a 150-mL conical flask, and 50 mL of 1 mol/L NH$_4$OAc was added for extraction in a shaker at 120 r/min for 30 min. After letting the mixture stand for 20 min, it was filtered and stored in a refrigerator at 4 °C for subsequent determination of AK. The content of AN and AP was measured using an AA3 continuous flow analyzer; the content of AK was measured using a flame photometer.

### 2.3.3. Total Carbon and Organic Matter Content

Total soil carbon (TC) and total organic carbon (TOC) were measured with a TOC-L total organic carbon analyzer [79,80]. Air-dried soil samples (0.05 g) were passed through a 0.15-mm steel sieve into the sample boat; they were then spread evenly and burned at 900 °C to determine the TC content. Soil samples in the sample boat were injected with H$_3$PO$_4$ and burned at 200 °C to determine the inorganic carbon (IC) content. The TOC content of soil was calculated by subtracting TC from IC. Total organic matter (TOM) was TOC multiplied by 1.724.

### 2.3.4. Soil pH

The pH of soil was measured using a pH meter [81]. Air-dried soil samples (20.0 g) were passed through a 1-mm steel sieve into a 50-mL beaker, and 20 mL of CO$_2$-free water was added, followed by stirring for 1 min to fully disperse the soil particles; they were then placed in a beaker for 30 min. The pH meter electrode was inserted into the tested liquid, and the beaker was shaken gently to remove the water film on the electrode; after the solution equilibrated, it was left to stand for a while, the reading switch was pressed, and the pH was recorded when the reading stabilized. The electrode was cleaned between measurements, and the positioning of the standard solution was assessed after every five to six samples.

### 2.4. Berry Analyses

### 2.4.1. Physicochemical Indexes of Grape Berries

A total of 100 grapes were randomly selected, and their weight was measured using an electronic balance. A subsample of 50 grapes was randomly selected, and juice from the grapes was manually extracted using gauze. The content of reducing sugar (RS, g/L) was determined using Fehling's reagent titration method; the content of titratable acid (TA, g/L) was determined using NaOH titration, and the soluble solids (SS) content was determined using a digital hand-held sugar meter [1]. The M value (the ratio of RS to TA) was calculated, which is the berry maturity coefficient and indicates the maturity of the berry.

### 2.4.2. Polyphenols Content in Grape Skins

The extraction of phenolic substances followed the procedure of [1,82] with slight modifications; polyphenols were extracted using a methanol solution of hydrochloric acid. Briefly, 100 grapes from those stored at $-80\ ^\circ$C were randomly selected, and their skins were peeled. Liquid nitrogen was added to the skins, and then the material was ground in a mortar to a fine powder. The powder was placed in a freeze dryer (FD-1C-50) for 24 h and then transferred to a plastic pack and stored in a freezer at $-20\ ^\circ$C. The phenolic constituents were then extracted with methanol-HCl (60% methanol, 0.1% HCl) and treated in an ultrasonic radiation machine using a ratio of 20 mL of solvent to 1 g of sample at $30\ ^\circ$C and 40 W for 30 min. The liquid extracts were separated from the solids by centrifugation (Eppendorf AG 22,331 Hamburg) at 10,000 rpm for 10 min, and the supernatants were collected and placed in glass bottles. All extractions were performed in triplicate, and the supernatants of the three extractions were collected, pooled, and stored in a freezer at $-20\ ^\circ$C. The above procedures were performed with protection from light.

The content of total phenols was determined following the procedure of [83], using the Folin–Ciocâlteu colorimetric method, but with slight modifications. First, 100 μL of the grape peel dry powder extract to be tested and 0.9 mL of water were added to a 20-mL glass test tube. Next, 5.00 mL of water was added; after the solution was shaken well, 0.2 mL of Folin–Ciocâlteu reagent was added, and the solution was shaken well. After 2 min, 2.0 mL of 10% sodium carbonate solution was added; the solution was mixed by shaking, and 0.9 mL of water was added. After 60 min of reaction in the dark, the absorbance was measured colorimetrically at a wavelength of 765 nm. Results were expressed in gallic acid equivalents.

The content of anthocyanins was measured following the procedure of [84], using the AOAC pH differential method, but with slight modifications. The grape peel dry powder extract was diluted 20 times with pH 1.0 hydrochloric acid-sodium chloride buffer and pH 4.5 acetic acid-sodium acetate buffer, and the absorbance of these two dilutions was then measured at 510 nm and 700 nm, respectively. Results were expressed in delphinidin-3-glucoside equivalents.

The content of flavan-3-ol was determined following the procedure of [85], using the p-DMACA-HCl method, but with slight modifications. Briefly, 0.1 mL of grape peel dry powder was added to a 10-mL glass test tube, followed by 3 mL of 0.1% p-DMACA in 1.0 mol/L hydrochloric acid methanol solution, and the solution was shaken, mixed well, and left to react at room temperature for 10 min. The absorbance was measured at 640 nm. The results were expressed in (+)-catechin equivalents.

The content of flavonoids was measured following the procedure of [86] using the $NaNO_2$-$AlCl_3$ spectrophotometric method, but with slight modifications. First, 0.3 mL of grape peel dry powder extract and 0.7 mL of methanol were added to the reaction tube; 2.7 mL of 30% methanol was added, and the solution was shaken well. Next, 0.2 mL of 0.5 mol/L sodium nitrite solution was added, and the solution was shaken well. This was followed by the addition of 0.2 mL of 0.3 mol/L aluminum chloride solution; after shaking the solution well and letting it stand for 5 min, 1 mL of 1 mol/L sodium hydroxide solution was added, and the solution was again shaken well. The absorbance was measured at 510 nm. Ultrapure water was used to replace the aluminum chloride solution as a background control. The results were expressed in rutin equivalents.

The content of tannins was determined based on the procedure of [87], using the methylcellulose precipitation method, but with slight modifications. First, 0.25 mL of grape peel dry powder extract was added to a 10-mL glass test tube, and 3 mL of methylcellulose solution was added to the sample group, which was inverted several times to mix the contents well. The solution was then left to stand for 2–3 min, and methylcellulose solution was not added to the control group. Two mL of saturated $(NH_4)_2SO_4$ solution was added to both the sample group and the control group, and the volume was adjusted to 10 mL with deionized water. Both the sample group and the control group were centrifuged at

$1800 \times g$, and the absorbance was measured at 280 nm. The absorbance value of tannin was obtained by subtracting the two values. The results were expressed in catechin equivalents.

*2.5. Statistical Analysis*

The experimental data were organized in Microsoft Office Excel 2017; all data were analyzed using IBM SPSS Statistics 21, and graphs were built using GraphPad Prism. Data were tested for normality and homogeneity of variance before one-way ANOVA. The *p*-values of the data are all greater than 0.05, suggest that the data are not statistically significant, obey the normal distribution, and have homogeneity of variance. For one-way ANOVA and Duncan's multiple comparison tests, the threshold for statistical significance was $p < 0.05$. The multiple comparison test was only used to make comparisons between treatments. Genescloud.cn (18 March 2022) was used to conduct the principal component analysis and correlation analysis.

## 3. Results

### *3.1. Soil Nutrients*

#### 3.1.1. Total Nutrients

The effects of in-row mulching on the total nutrients of the soil in 2019 and 2020 are shown in Figure 3. The TN content of the PF treatment was significantly higher than that of the LF treatment in the 0–20 cm soil layer in 2019; the TN content of the PF treatment was higher than that of the other treatments in 2020, although no significant differences in the TN content between treatments were observed (Figure 3a). There were no significant differences in the TP content of each treatment in the 0–20 cm layer in 2019 and 2020 (Figure 3c). There were no significant differences in the TK content of each treatment in the 0–20 cm layer in 2019; however, significant differences between the treatments were detected in the 0–20 cm layer in 2020. The TK content was 7.46% and 4.52% higher in the LF and PF treatments than in the CK, respectively, and these differences were significant (Figure 3e). There were no significant differences in the content of TN (Figure 3b), TP (Figure 3d), and TK (Figure 3f) in the 20–40 cm soil layer between the treatments.

#### 3.1.2. Available Nutrients

The effects of in-row mulching on the available soil nutrients in 2019 and 2020 are shown in Figure 4. In the 0–20 cm soil layer, the AN content of the PF treatment was significantly higher than that of the other treatments (Figure 4a), which was consistent with the patterns observed for the TN content. The soil AP content of each treatment was similar for both years of the experiment. Mulching increased the soil AP content, and the LF treatment had a more pronounced effect than the PF treatment. The AP content was 66.62% and 173.63% higher in the PF and LF treatments in 2019, and 88.60% and 209.89% higher in 2020, respectively, relative to CK (Figure 4c). The AK content of the LF treatment was significantly higher than that of the other treatments, and the AK content of the PF treatment was significantly higher than that of CK in 2019 (Figure 4e). There were no significant differences in the content of AN (Figure 4b), AP (Figure 4d), and AK (Figure 4f) between the treatments in the 20–40 cm soil layer.

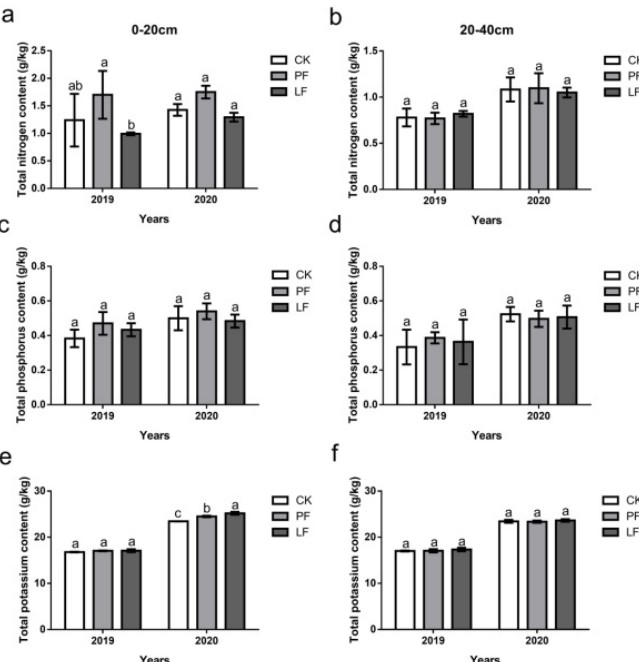

**Figure 3.** Effects of in-row mulching on the total nutrients of the soil in 2019 and 2020. (**a**): Total nitrogen content (0–20 cm); (**b**): Total nitrogen content (20–40 cm); (**c**): Total phosphorus content (0–20 cm); (**d**): Total phosphorus content (20–40 cm); (**e**): Total potassium content (0–20 cm); (**f**): Total potassium content (20–40 cm). Notes: Values are the mean $\pm$ SD of three biological replicates. Different letters above the columns indicate significant differences between treatments in Duncan's multiple comparisons ($p < 0.05$).

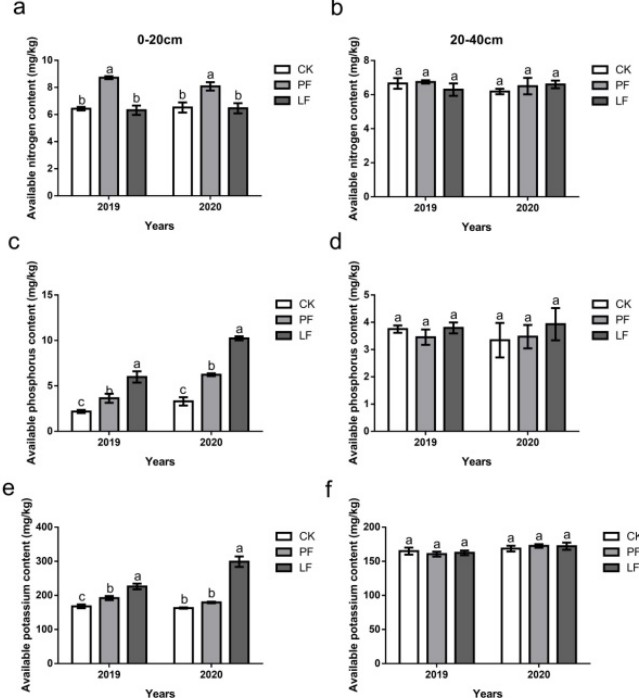

**Figure 4.** Effects of in-row mulching on the available nutrients of the soil in 2019 and 2020. (**a**): Available nitrogen content (0–20 cm); (**b**): Available nitrogen content (20–40 cm); (**c**): Available phosphorus content (0–20 cm); (**d**): Available phosphorus content (20–40 cm); (**e**): Available potassium content (0–20 cm); (**f**): Available potassium content (20–40 cm). Different letters above the columns indicate significant differences between treatments in Duncan's multiple comparisons ($p < 0.05$).

### 3.1.3. Total Carbon and Organic Matter Content

There were no significant differences in the TC and TOM content between treatments within any year (2019 or 2020), or soil layer (0–20 cm and 20–40 cm); however, the TC and TOM content was higher in 2019 than in 2020 (Figure 5).

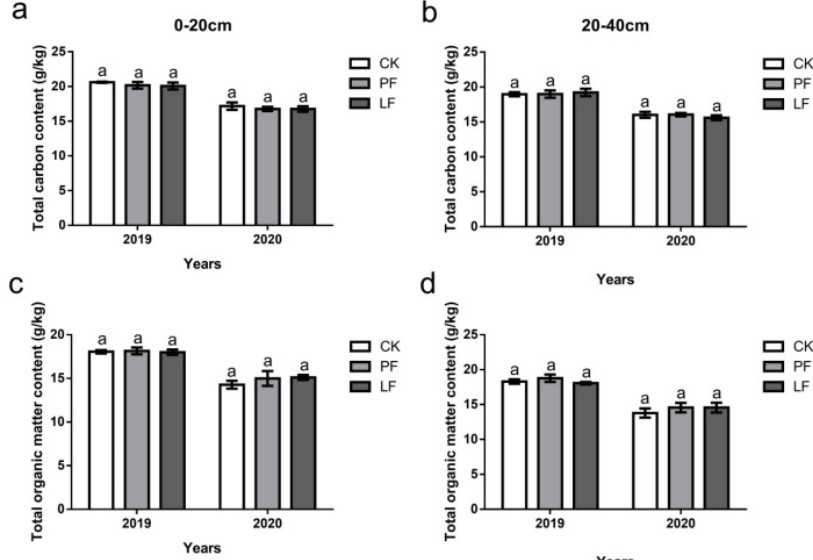

**Figure 5.** Effects of in-row mulching on the total carbon and organic matter contents of the soil in 2019 and 2020. (**a**): Total carbon content (0–20 cm); (**b**): Total carbon content (20–40 cm); (**c**): Total organic matter content (0–20 cm); (**d**): Total organic matter content (20–40 cm). Different letters above the columns indicate significant differences between treatments in Duncan's multiple comparisons ($p < 0.05$).

### 3.1.4. Soil pH

The pH of the LF treatment was significantly lower than that of the other treatments in the 0–20 cm soil layer, and there was no significant difference in the pH between treatments in the 20–40 cm layer (Figure 6).

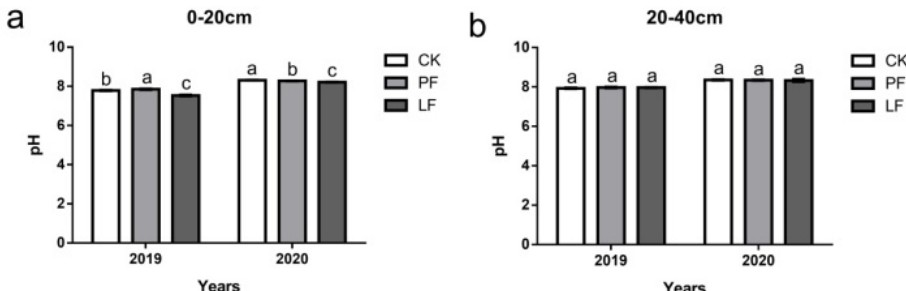

**Figure 6.** Effects of in-row mulching on the pH of the soil in 2019 and 2020. (**a**): pH (0–20 cm); (**b**): pH (20–40 cm). Different letters above the columns indicate significant differences between treatments in Duncan's multiple comparisons ($p < 0.05$).

### 3.1.5. Principal Component Analysis

A principal component analysis (PCA) was performed to visualize the effects of covering treatment on soil nutrients, and the results are shown in Figure 7. The first three principal components (PCs) accounted for 89.5% of the variation in soil nutrients (PC1: 52.0%; PC2: 24.7%; and PC3: 12.8%). Clear separation was observed between the different samples (2019CK, 2019PF, 2019LF, 2020CK, 2020PF, and 2020LF) (Figure 7a). The PC1–PC2 plane is shown in Figure 7b. The Euclidean distances between the 2019CK, 2019PF, and

2019LF samples were small, followed by the distances between the 2020CK, 2020PF, and 2020LF samples; the Euclidean distances between 2019CK and 2020CK, between 2019PF and 2020PF, and between 2019LF and 2020LF were relatively large. These findings indicated that the differences stemming from covering treatment were minor compared with the differences between the two years. The distances between the samples for each treatment in 2019 differed from those in 2020, suggesting that the effect of covering on soil nutrients may differ between vintages.

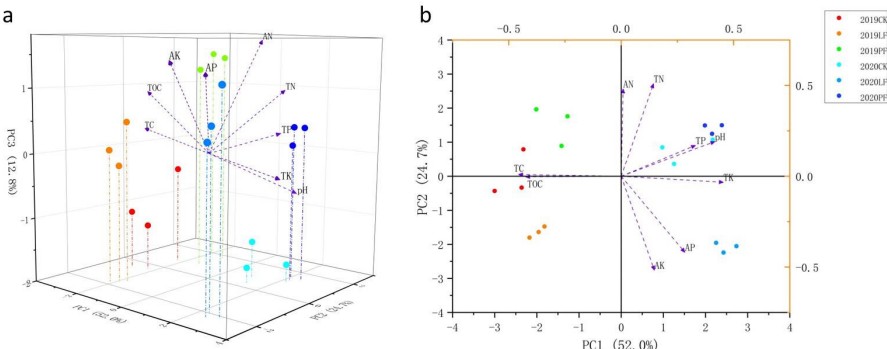

**Figure 7.** Principal component Analysis (PCA) of soil property index in 2019 and 2020. (**a**): Stereogram including PC1, PC2 and PC3; (**b**): Floor plan including PC1 and PC2.

The TC and TOM content was closer to the 2019 CK samples, and contributed the most to variation in soil nutrients among the 2019 CK samples. Similarly, the TC, TOM, TN, and AN content contributed the most to variation in soil nutrients among the 2019 PF samples. The TC, TOM, AP, and AK content contributed significantly to variations in soil nutrients among the 2019 LF samples. The pH contributed significantly to variations in soil nutrients among the 2020 CK samples. The TN, TP, and AN content contributed heavily to variations in soil nutrients among 2020 PF samples. The TK and AP content contributed heavily to variations in soil nutrients among the 2020 PF samples.

*3.2. Quality of Berries*

3.2.1. Physicochemical Indexes of Grape Berries

The effects of in-row mulching on the physicochemical indexes of grape berries after ripening in 2019 and 2020 are shown in Figure 8. The 100-berry weight of grapes in each treatment was consistent within each of the two years; the 100-grain weight of the PF treatment was significantly higher than that of CK, and there was no significant difference in the 100-grain weight between the LF treatment and CK (Figure 8a). The SS content of the mulching treatment was higher than that of CK in both 2019 and 2020. The difference was that the SS content of the LF treatment was significantly higher than that of the PF treatment in 2019, but there was no significant difference in the SS content between these two treatments in 2020 (Figure 8b). The mulching treatments increased the titratable acid content of grape berries in both 2019 and 2020, and the titratable acid content of each treatment was higher in 2020 than in 2019. The titratable acid content of the PF treatment significantly differed from that of CK in both years; the titratable acid content of the PF treatment only significantly differed from that of CK in 2019 (Figure 8c). Both mulching treatments significantly increased the reducing sugar content of grape berries, and the reducing sugar content of the LF treatment was significantly higher than that of the PF treatment in 2020 (Figure 8d). Mulching treatments significantly increased the M values in grape berries, and the M values were significantly higher in the PF treatments than in the LF treatments in 2019; the opposite pattern was observed in 2020 (Figure 8e).

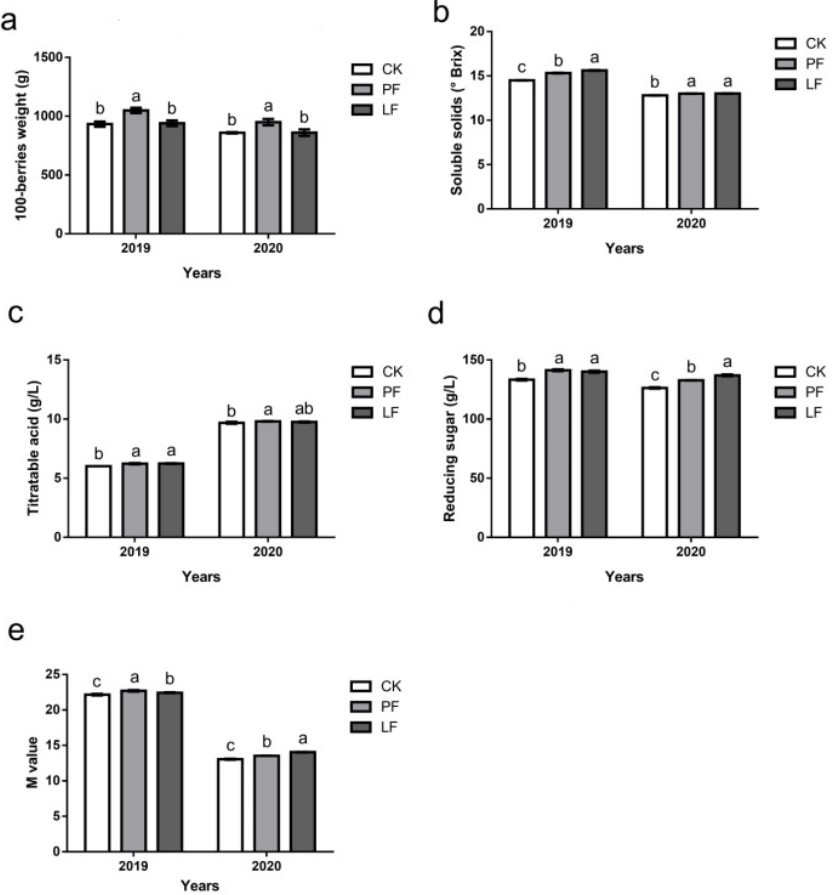

**Figure 8.** Effects of in-row mulching on the physicochemical indexes of grape berries in 2019 and 2020. (**a**): 100-grain weight; (**b**): Soluble solids (SS); (**c**): Titratable acid (TA); (**d**): Reducing sugar; (**e**): M value. Different letters above the columns indicate significant differences between treatments in Duncan's multiple comparisons ($p < 0.05$).

3.2.2. Polyphenols in Grape Skin

The polyphenol content in grape skin was determined in ripe grapes harvested with different mulching treatments, and the results are shown in Figure 9. The total phenolic content of each treatment was consistent in 2019 and 2020; it was higher in the mulching treatment than in CK, and significantly higher in the LF treatment than in the PF treatment (Figure 9a). The total anthocyanin content of each treatment was consistent in 2019 and 2020; the total anthocyanin content was significantly higher in the LF and PF treatment than in CK, and no significant difference in the anthocyanin content was observed between the LF and PF treatments (Figure 9b). Ground cover increased the flavan-3-ol content in grape skin; no significant difference between the PF and CK treatments was observed in the flavan-3-ol content, but the content of flavan-3-ol was significantly lower in the PF and CK treatments than the LF treatment in 2019. Furthermore, the content of flavan-3-ol was significantly higher in the LF and PF treatments than in CK; it was also significantly higher in the LF treatment than in the PF treatment in 2020 (Figure 9c). The flavonoid content of each treatment was similar in 2019 and 2020. The flavonoid content was significantly higher under the mulching treatment than in the CK, and it was also significantly higher in the LF treatment than in the PF treatment (Figure 9d). Ground cover increased the tannin content in grape skin, and the increase was higher in the LF treatment than in the PF treatment in 2019, and higher in the PF treatment than in the LF treatment in 2020. (Figure 9e)

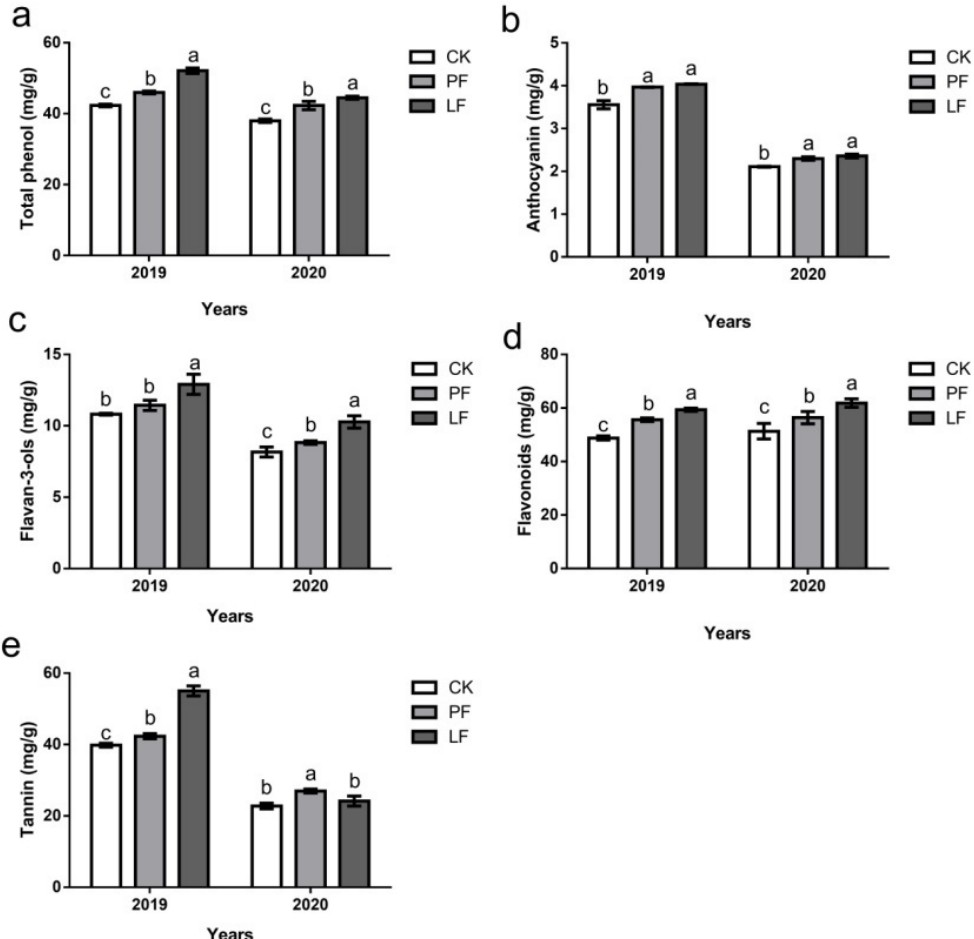

**Figure 9.** Effects of in-row mulching on the content of polyphenols in grape skins in 2019 and 2020. (**a**): Total phenol; (**b**): Anthocyanin; (**c**): Flavan-3-ols; (**d**): Flavonoids; (**e**): Tannin. Different letters above the columns indicate significant differences between treatments in Duncan's multiple comparisons ($p < 0.05$).

3.2.3. Principal Component Analysis

PCA was performed to visualize the effects of covering treatment on berry quality, and the results are shown in Figure 10. The first two PCs accounted for 90.7% of the variation in berry quality (PC1: 73.2%; PC2, 17.5%). Clear separation was observed between the different samples (2019CK, 2019PF, 2019LF, 2020CK, 2020PF, and 2020LF) (Figure 10). The Euclidean distances between treatments in the same year were short, followed by the distance between the same treatments in different years. The results indicated that the differences stemming from cover treatment were minor compared with the differences between these two years. The distances between the samples for each treatment in 2019 were different from those in 2020, suggesting that the effect of ground cover on berry quality might differ between vintages.

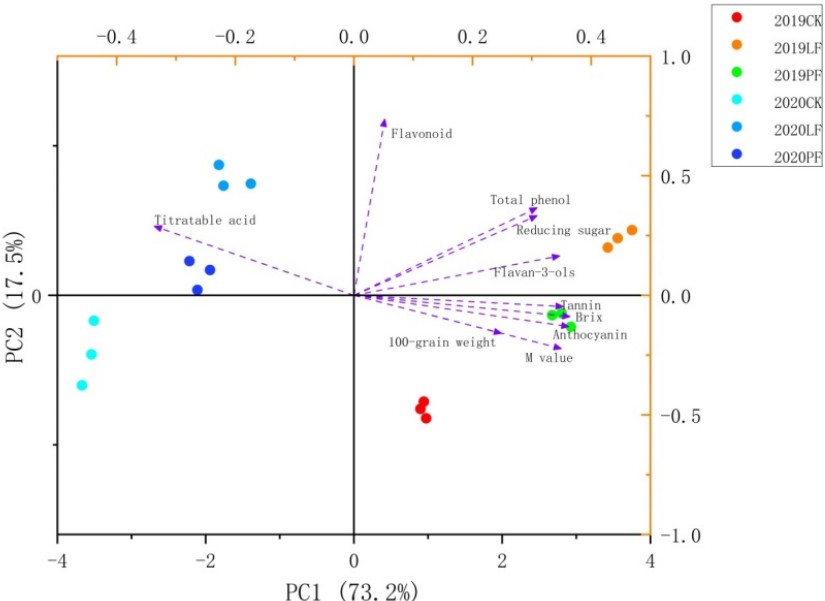

**Figure 10.** Principal component Analysis (PCA) of fruit quality index in 2019 and 2020.

Most compounds, with the exception of titratable acid and flavonoids, were close to 2019 samples and contributed heavily to the fruit quality of 2019 samples. Similarly, the titratable acid content contributed significantly to the fruit quality in 2020. Flavanols also contributed significantly to 2020 LF samples.

*3.3. Correlation Analysis between Soil Properties and Fruit Quality*

A correlation analysis was conducted to evaluate the relationships between soil properties and fruit quality indicators (Figure 11). The content of reducing sugar, flavonoid, total phenol, M value, flavan-3-ols, and anthocyanin was significantly positively correlated with the TK, AP, and AK content, but significantly negatively correlated with pH. In addition, the Brix value was significantly positively correlated with TK and AP, and negatively correlated with pH. The content of titratable acid was significantly positively correlated with TP and negatively correlated with TC. The anthocyanin content was significantly positively correlated with TOM.

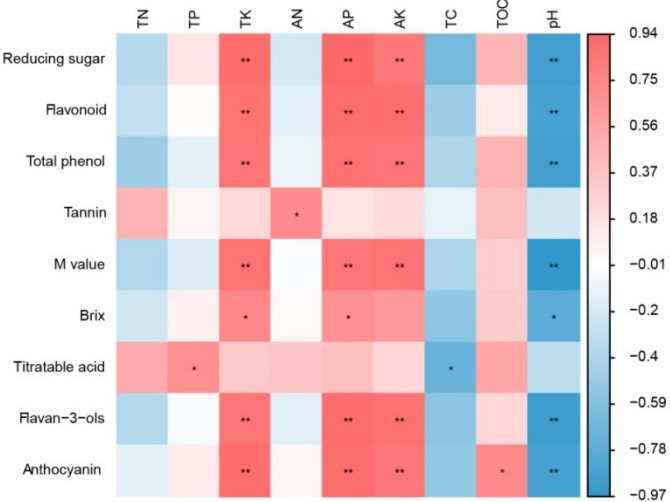

**Figure 11.** Correlation analysis between soil property and fruit quality. Note: * means the two indicators are significantly different at the 0.05 level ($p < 0.05$); ** means the two indicators are extremely significantly different at the 0.01 level ($p < 0.01$).

## 4. Discussion

N, P, and K are three essential mineral nutrients that are in high demand for grape growth [88]. They alter the growth of fruit trees through physiological or biochemical metabolic processes and also affect the growth and development, yield, and quality of fruit trees [89]. N, P, and K in plants are mainly derived from the soil. Therefore, the study of soil nutrition is important for understanding the growth of vines. The results of this study showed that the content of TN, TP, and TK in the mulching treatments in the 0–20 cm and 20–40 cm soil layers was not significantly different from that of the CK in 2019 and 2020 (Figure 3), with the exception of TK in 2020. This indicates that mulching had little effect on the total amount of N, P, and K elements in soil within each of the two years. Ground cover can improve the effectiveness of N, P, K fertilizer application. PF significantly increased the AN content in the 0–20 cm soil layer, and the two mulching treatments had similar effects on the content of AP and AK in 2019 and 2020; all showed that the content of AP and AK after mulching was significantly higher than that of CK, and the LF treatment had a more significant effect (Figure 4). This is basically consistent with the results of [31,34], but inconsistent with the results of [90]. This might be explained by differences in experimental conditions, as well as the fact that mulching measures have different effects on soil nutrients. In addition, long-term plastic film mulching deteriorates the soil's physical and chemical properties [36,37] and continuously reduces the soil's organic carbon content [38]. However, long-term surface mulching with straw can keep soil microbial growth and reproduction in a vigorous state, resulting in an increase in soil organic carbon content year by year [91]. Wang Y. suggested that organic mulching practices increase soil nutrients, whereas inorganic mulching practices did the opposite. This is consistent with the main goal of this experiment (demonstrating the efficacy of LF). Therefore, LF can be used as a substitute material for PF. Mulching can increase the soil nutrient content to varying degrees, probably because mulching improves soil structure, hydrothermal conditions, and microbial activity. Soil organic matter plays a key role in determining soil function and quality [92], and a high organic matter content can increase the soil nutrient supply [93], improve soil physicochemical properties and microbial activity, and improve the soil buffering capacity [94]. TOM plays an important role in maintaining soil fertility and the productivity of agro-ecosystems; therefore, increasing TOM is the main goal of using organic mulch. Numerous studies have shown that organic mulching can increase soil biomass carbon input, thereby increasing TOM content of the soil [95,96]. In our study, the content of TC and TOM did not significantly differ between the three treatments within each of the two years, indicating that there was no significant change in the C content of each soil layer (Figure 5), which might stem from the low number of years of our study, as well as the fact that the accumulation of organic matter is not achieved deep in the soil. The TC and TOM content was higher in 2019 than in 2020, indicating that the soil C content differed between years. This might be caused by climatic differences between years. The roots of grapevines are generally distributed in the soil at a depth of 15–80 cm, mostly concentrated at 20–40 cm [3–6]. The growth of roots is water-oriented, fertilizer-oriented, and geotropic. The deeper the fertilization, the deeper the root system is, and vice versa [7–9]. This experimental site watered by drip irrigation is an organic vineyard without fertilizers and pesticides. The luxuriant growth of shallow and lateral roots is the possible reason why the improvement of soil quality in the 0–20 cm layer affected the quality of grape fruit in this experiment. However, this has not been confirmed. In this experiment, there was no significant difference in the soil quality index between 20–40 cm for each treatment, indicating that the two years of mulching did not affect the soil layer where the deep roots were located, but only affected the soil near the shallow roots. In sum, mulching LF can improve the availability of surface soil nutrients, and the climatic differences between different years, the rainfall, and the mulching year also affect the content of soil nutrients. In grapes, the content of sugars accounts for approximately 25% [97], which are precursors for the synthesis of pigments, vitamins, and some volatile aroma substances [98,99]. Studies have shown that in the

process of grape ripening, polyphenols increase with the sugar content. After the sugar content is increased to a sufficient level, the content of polyphenols and aroma substances also peaked [100]. Similar to sugars, acids are also important taste substances in grapes, which can reflect the sensory characteristics of grapes. Acidic substances can promote the dissolution of pigment substances in grape skins and play an important role in the extraction of pigment substances from grape skins. The sugar–acid ratio (M value) generally indicates the maturity of grape [101]. Ground cover can increase the ground temperature in the low-temperature season, promote the growth and maturation of grapes and nutrient transformation, preserve moisture, and conserve water, which increases the size, quality, and SS content of grape fruits, and thus grape quality [102–104]. A previous mulching study on Yuluxiang pear showed that the single fruit quality, pulp hardness, and SS content were the highest in the ground-cloth mulching treatment [105]. Ground cover can also increase the chlorophyll content of grape leaves [34,35], and then accelerate changes in the color of grapes by enhancing photosynthesis and improving fruit maturity [31], thereby increasing fruit yield and quality. Our findings support this perspective. The results of this study showed that PF coverage significantly increased the 100-berry weight of grape berries, and there was no significant difference in the 100-berry weight between LF and CK. The two mulching treatments significantly increased the SS, reducing sugar, and titratable acid content of grape berries in both 2019 and 2020. The increase in the reducing sugar content under mulching treatment was larger than that of the titratable acid content relative to CK, which increases the ripening coefficient (M value). This indicates that in-row mulching can improve the ripeness of the grape fruit.

Phenolic substances are one of the most important quality components of grape fruit [106] and an important component of the wine skeleton, which determines the astringency, bitterness, and antioxidant properties of grapes and wines [107]. Phenolic substances mainly include anthocyanins, procyanidins, flavanols, flavonols, and tannins [108]. Anthocyanins generally exist in the skin of red grapes, provide color to the wine in the process of brewing wine, and determine the appearance quality of red wine [109]; however, its stability is affected by temperature and pH [110]. Proanthocyanidins have anti-aging effects and prevent cardiovascular and cerebrovascular diseases [111], as they are one of the most effective free radical scavengers [112]. Flavanols and flavonols in wine affect the health benefits provided by wine consumption. Flavonols only exist in grape skins, and flavanols are present in grape skins, seeds, and stems. They are also bitter [113]. The main tannin in wine is condensed tannin, a compound formed by the polymerization of flavan-3-ol monomers in grape skins and seeds; it has a strong antioxidant capacity, which makes wine age [114–117]. Previous studies have shown that ground cover can increase the chlorophyll content of grape leaves, induce changes in grape color, and improve fruit quality and yield by enhancing photosynthesis [34,35]. Orchard grass mulch can also increase the content of phenolics in berries [74]. This might be related to the improvement of the light conditions in the middle and lower parts of the orchard by ground cover [118]. The results of this study showed that mulching could significantly increase the content of total phenols, anthocyanins, flavan-3-ols, flavonoids, and tannins in grape skins. These findings are basically consistent with the results of previous studies [20,31,34]. However, the effects of mulching on plant physiology, especially photosynthesis, require increased attention in future studies.

Soil nutrient status is closely related to plant growth. The organic matter in the soil is the source of soil nutrient elements. Microbes decompose soil organic matter and minerals and release nutrient elements for plant absorption and utilization [119]. Previous studies have shown that a higher nitrogen content can promote the growth of grapes and the accumulation of aromatic substances in grapes, and potassium and phosphorus elements can promote the growth of flowers and fruits [120]. The results of our experimental study supported these speculations, as the content of reducing sugars, flavonoids, total phenols, M value, flavan-3-ols, and anthocyanins were significantly positively correlated with TK, AP, and AK. An in-depth analysis of the changes in soil structure and soil microorganisms

after mulching is needed. The effects of LF and PF treatments on soil and fruit quality were basically the same (Figures 3–6, 8 and 9). Given that LF is more environmentally friendly, LF has greater application potential compared with PF.

## 5. Conclusions

The aim of this study was to characterize the effects of LF, a new type of mulching material, on soil and grape berry quality compared with PF and CK. The results revealed that mulching can increase the availability of N, P, and K in the surface (0–20 cm) soil but has no significant effect on the total nutrient and C content. Mulching treatment significantly increased the ripeness of the grape berries, as well as the content of phenolics in the skins. There was a significant correlation between soil nutrients and grape fruit quality, and the correlations with TK, AP, and AK were particularly important. However, soil and fruit quality varied between years due to variations in rainfall and temperature. The differences in climate, rainfall, and mulching between years also affected the content of soil nutrients. LF increased the content of AP, AK, total phenols, flavan-3-ols, and flavonoids over PF in the same year. Thus, our findings indicate that LF is an effective substitute for PF.

**Author Contributions:** Conceptualization, X.D., Y.Y. and X.H.; methodology, Y.W.; software, X.D.; validation, R.L., F.G. and L.Z.; formal analysis, X.D. and R.W.; investigation, Y.Y.; resources, X.D. and Y.Y.; data curation, X.D.; writing—original draft preparation, X.D.; writing—review and editing, X.H. and Y.W.; visualization, X.D.; supervision, H.W.; project administration, H.W.; funding acquisition, H.L. All authors have read and agreed to the published version of the manuscript.

**Funding:** This research was funded by Ningxia Hui Nationality Autonomous Region Major Research and Development Project—Research and demonstration on key technology of wine style curing in the eastern foot of Helan Mountain in Ningxia, grant number 2020BCF01003 and Research and application of key technologies for sustainable development of wine industry, grant number LYNJ202110.

**Institutional Review Board Statement:** Not applicable.

**Informed Consent Statement:** Not applicable.

**Data Availability Statement:** The data presented in this study are available in this article.

**Acknowledgments:** We would like to thank the Shaanxi Kerui Company for providing the biodegradable liquid film.

**Conflicts of Interest:** The authors declare no conflict of interest. The funders had no role in the design of the study; in the collection, analyses, or interpretation of data; in the writing of the manuscript, or in the decision to publish the results.

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
