# Peer review of "Effects of Biodegradable Liquid Film on the Soil and Fruit Quality of Vitis Franco-american L. Hutai-8 Berries"

_horticulturae, doi:10.3390/horticulturae8050418_

Round 1

Reviewer 1 Report

The manuscript is very interesting and the topic is relevant regarding the environmentally friendly materials in horticulture, and preserving and enhancing soil nutrients and humidity.

Some comments:

The introduction part should include some facts of grape rooting system - how deep and in which soil layer depths are the most roots, and from which depth are the nutrients the most available for grapes. Moreover, this can later be integrated into the discussion part in relation to the content of nutrients in the 0-20 cm and 20-40 cm soil layers. The author have mentioned in only very briefly in the discussion part.

And if the major part of the grape roots are in deeper soil levels, then how the mulch really affect the roots? It is logical that the ground cover mulching affects only the upper layer of soil, but as the grapevine roots can go into very deep layers of soil - the impact of mulching on nutritional status should be more carefully considered. Mostly in long-term crops, the mulch is used more for weed control and humidity purposes. Two years for such experimental effects is a bit too short, but it is possible to show preliminary results.

Did you check the root formation in any grapevine plants in different mulch treatments? 

Row 292 - Add another row spacing before the next sub-paragraph.

Row 348, 480, 481 - The term 100-grain weight is incorrect for grapes, it should be 100-berry weight - please correct it. In the figure 7 it is used correctly.

Row 424 - Revise the sentence in this row. (emainly?)

Please format the references by the standards required by the MDPI Horticulturae, and check the other formatting flaws as well.

Reviewer 2 Report

Covering the soil of vineyards has become a focus of interest again in the context of climate change. The subject of the manuscript under review is likely to be of broad interest.
The manuscript is well-edited and meets MDPI standards. The use of English terminology could be slightly improved, and I suggest that the authors follow the terminology commonly used in the viticultural literature (e.g. orchard instead of vineyard, fruit tree instead of stock, ear instead of bunch).
Is the meteorological data in Figure 1 their own measurements, or is it also taken from the China Statistical Yearbook? Please indicate.
A more detailed description or reference of the cultivar studied would be good.
For the experimental setup, I suggest that a figure be prepared and included in the manuscript.
What color is the LF mulch? Please indicate. This might influence the effect.
It would be good to insert a reference on the berry sampling.
Chapter 2.5: Have you checked the ANOVA assumptions? What test? Please describe.
For the introduction and discussion: mulching can have negative effects, e.g. pentosan effect, which can make mechanical cultivation more difficult due to the occasional slippage hazard. Highlight these.
